# Genomic structure and transcript analysis of the Rapid Alkalinization Factor (RALF) gene family during host-pathogen crosstalk in *Fragaria vesca* and *Fragaria* x *ananassa* strawberry

**Francesca Negrini**[1,2], **Kevin O'Grady**[2¤], **Marko Hyvönen**[3], **Kevin M. Folta**[2], **Elena Baraldi** [1] *

**1** Laboratory of Plant Pathology and Biotechnology, DISTAL, University of Bologna, Bologna Italy,
**2** Horticultural Sciences Department, University of Florida, Gainesville, Florida, United States of America,
**3** Department of Biochemistry, University of Cambridge, Cambridge, United Kingdom

¤ Current address: Department of Microbiology and Cell Science, University of Florida, Gainesville, Florida, United States of America

* elena.baraldi@unibo.it

**Data Availability Statement:** All relevant data are within the paper and its Supporting Information files.

## Abstract

Rapid Alkalinization Factors (RALFs) are cysteine-rich peptides ubiquitous within plant kingdom. They play multiple roles as hormonal signals in diverse processes, including root elongation, cell growth, pollen tube development, and fertilization. Their involvement in host-pathogen crosstalk as negative regulators of immunity in *Arabidopsis* has also been recognized. In addition, peptides homologous to RALF are secreted by different fungal pathogens as effectors during early stages of infection. Previous studies have identified nine RALF genes in the diploid strawberry (*Fragaria vesca*) genome. This work describes the genomic organization of the RALF gene families in commercial octoploid strawberry (*Fragaria × ananassa*) and the re-annotated genome of *F. vesca*, and then compares findings with orthologs in *Arabidopsis thaliana*. We reveal the presence of 15 RALF genes in *F. vesca* genotype Hawaii 4 and 50 in *Fragaria x ananassa* cv. Camarosa, showing a non-homogenous localization of genes among the different *Fragaria x ananassa* subgenomes. Expression analysis of *Fragaria x ananassa* RALF genes upon infection with *Colletotrichum acutatum* or *Botrytis cinerea* showed that *FanRALF3-1* was the only fruit RALF gene upregulated after fungal infection. *In silico* analysis was used to identify distinct pathogen inducible elements upstream of the *FanRALF3-1* gene. Agroinfiltration of strawberry fruit with deletion constructs of the *FanRALF3-1* promoter identified a 5' region required for *FanRALF3-1* expression in fruit, but failed to identify a region responsible for fungal induced expression.

**Funding:** Protein studies were conducted at the University of Cambridge, UK, thanks to EMBO short fellowship number 7779. FN has been supported by PhD fellowship from MIUR (Ministry of Instruction, University and Research). The funders had no role in study design, data collection and analysis, decision to publish, or preparation of the manuscript.

**Competing interests:** The authors have declared that no competing interests exist.

## Introduction

In plants, several small secreted peptides (SSPs) function as hormonal signalling molecules that respond to internal and external stimuli [1]. SSPs are known to be involved in different processes, ranging from organ growth to biotic and abiotic responses [2, 3].

Rapid alkalinization factors (RALFs) are cysteine-rich SSPs originally identified for their ability to rapidly alkalinize tobacco cell culture [4]. They are ubiquitous in the plant kingdom with 37 members identified in *Arabidopsis thaliana* genome alone [5, 6]. RALF genes are translated as pre-pro-proteins and are activated in the apoplast through proteolytic cleavage, generating ~50 amino acid long active peptides. Aside from the signal sequence necessary for extracellular extrusion, canonical RALF peptides contain distinctive amino acid motifs, such as the RRILA motif for S1P protease recognition [7] and the YISY motif, important for the activation of the signaling cascade [8, 9]. In addition, four conserved cysteines form two disulfide bonds that stabilise the mature RALF peptide. Based on these features, RALFs have been grouped into four major clades [6]; clades I, II and III contain typical RALF peptides, whereas clade IV includes the most divergent RALF proteins lacking, both the RRILA and YISY conserved motifs and in some cases containing only three of the four cysteines.

RALF peptides bind to the *Catharanthus roseus* Receptor Like Kinases 1—like family proteins (*Cr*RLK1L) known to be involved in cell expansion and reproduction throughout the plant kingdom [10]. The large *Cr*RLK1L receptor family which includes FERONIA (FER) proteins, previously were reported to interact with *Arabidopsis RALF1* and *RALF23* [11, 12], BUDDHA'S PAPER SEAL 1 and 2 (BUPS1/2), ANXUR1, 2 (ANX1/2) proteins interact through their ectodomain and bind to RALF4 and 19 in the pollen tube [13] and THESEUS1 (THE1) the RALF34 receptor in the root [14]. Binding RALF peptides to *Cr*RLK1L receptors also involves other interacting partners such as Lorelei-like-Glycosylphosphatidylinositol-Anchored proteins (LLG1,2,3) [9, 15] and Leucine-Rich Repeat Extensins (LRX). The latter has been reported to bind RALF4/19 in the pollen tube [16, 17] and to interact with FER, as part of cell wall sensing system responsible for vacuolar expansion and cellular elongation [18]. Binding of RALFs to their receptors leads to a number of different intracellular signaling events involving different molecular components, mostly still unidentified. However, it is known that in *A. thaliana* binding of RALF1 to FER receptor results in the phosphorylation of plasma membrane H(+)-adenosine triphosphatase 2, inhibition of proton transport and subsequent apoplastic alkalinization [11].

RALF peptides regulate a variety of different functions such as cell expansion [11], root growth, root hair differentiation [19, 20, 21] stress response [22], pollen tube elongation and fertilization [13, 16, 22, 23]. In *A. thaliana*, RALF peptides can also act as negative regulators of the plant immune response following bacterial infection [12]. For example, RALF23 binding to the FER receptor inhibits physical interaction between the immune receptor kinases EF-TU RECEPTOR (EFR) and FLAGELLIN-SENSING 2 (FLS2) with their co-receptor BRASSINOSTEROID INSENSITIVE 1–ASSOCIATED KINASE 1 (BAK1), inhibiting immune signalling. Interestingly, biologically-active RALF homologs have also been identified in fungal plant pathogens, possibly following horizontal gene transfer, suggesting a role for fungal RALF genes as virulence factors [24]. In fact, a fungal RALF homolog is required for host alkalinisation and infection by *Fusarium oxysporum* [25]. Because alkalinisation is important for activation of virulence factors and successfully infection of plant tissues for many pathogenic fungi, secreted RALF peptides may promote host alkalinisation at early stage infection when hyphal biomass is not sufficient to secrete a large quantities of ammonia [24]

Strawberry is an important crop but is susceptible to fungal and bacterial pathogens that can have a significant effect on marketable yield [26]. Post-harvest molds are particularly

difficult to manage as infection initiates during flowering, and can become problematic after a long asymptomatic quiescent period on ripe fruits [27]. Severe strawberry fruit post-harvest molds are caused by *Colletotrichum acutatum*, causal agent of anthracnose disease, and *Botrytis cinerea*, causal agent of grey mold [28].

*F. × ananassa* fruits infected with *C. acutatum* show increase in the accumulation of a RALF transcripts in the ripe susceptible fruit at an early stage of infection [29]. Overexpression of the *F. × ananassa* ortholog of *A. thaliana RALF33*, through transient agroinfiltration, increases strawberry fruit susceptibility to anthracnose, leading to increased fungal growth on fruits and the induction of the host immune response [30, 31]. Upregulation of RALF transcripts during plant infection has also been observed in mature red tomato fruits (*Solanum lycopersicum*) upon infection with *Colletotrichum gleosporioides* [32] and in rice (*Oryza sativa)* upon *Magnaporthe oryzae* infection [31]. These findings suggest a role for RALF gene expression as a susceptibility factor in fungal infection. Furthermore Dobón *et al.* [33], studying the expression pattern of four *Arabidopsis* transcription factors mutants (*at1g66810*, *pap2*, *bhlh99*, *zpf2*) with increased susceptibility to *B. cinerea* and *Plectosphaerella cucumerina*, observed a coincidental increase of *RALF23*, *RALF24*, *RALF32* and *RALF33* transcripts.

The woodland strawberry RALF gene family members have been previously characterized based on the *F. vesca* genome annotation v1 [6], and nine *FveRALF* genes were identified and grouped in the four RALF clades. Analysis of an updated and annotated version of the *Fragaria vesca* genome (v4.0.a2) [34] and of the *Fragaria x ananassa cv*. Camarosa genome sequence (v1.0.a1) [35, 36], allowed exploration of RALF gene family composition and genomic organization. Genome localization, phylogenetic and transcript analyses were conducted, based on available genomic and transcriptomic sources, with an aim to gain insights into the functional roles of specific RALF genes. Furthermore, the induction of RALF gene expression upon infection of *F.× ananassa* fruits with *C. acutatum* and *B. cinerea* was studied. *In silico* analysis of the *FanRALF3-1* promoter was conducted in order to identify putative pathogen responsive motifs. We then tested *in vivo* if progressively truncated *FanRALF3-1* promoter fragments could induce reporter genes expression in agroinfiltarted strawberry fruits infected with *C. acutatum*.

## Materials and methods

### Identification of RALF family genes, phylogenetic analysis, and chromosome assignment

*F. vesca* RALF genes identified in v1.1 genome and reported by Campbell and Turner [6] were detected in the Genome Database for Rosaceae (GDR; v4.0.a2) through keyword gene search 'RALF' and through a BLASTn search using the previously annotated RALF sequences against the *F. vesca* v4.0.a1 chromosome database. The v4.0.a2 [34] and the Blastn gene search outputs were then used to update the previous data on v1.1 genome. Nucleotide sequences of the 15 *F. vesca* RALF members (both previous and new found ones), were used as query for BLASTx on the *F. × ananassa cv*. Camarosa Genome v1.0.a1 proteome [37] to find octoploid RALF orthologs. Sequences of *F. × ananassa* RALF peptides were aligned with MUSCLE [38] and the phylogeny was inferred using the Maximum Likelihood method and JTT matrix-based model [39]. The tree with the highest log-likelihood was chosen. The Initial tree was obtained automatically by applying Neighbor-Join and BioNJ algorithms to a matrix of pairwise distances estimated using a JTT model, and then selecting the topology with a superior log-likelihood value. Evolutionary analyses were conducted in MEGA X [40]. To define the classification of the new members in different clades, all the RALF protein sequences available from the Campbell and Turner annotation *F. vesca* genes were aligned with the newly identified RALFs and a

phylogenetic tree was constructed as mentioned above. *Fragaria x ananassa* genes annotation and position were retrieved from GDR (*F. × ananassa* cv. Camarosa genome v1.0.a1), and progenitor lineage were inferred according to [36]. Chromomap package in R [41] was used to create a *FanRALF* gene chromosome map.

## Expression profile of RALF family genes in *F. vesca*

Transcript accumulation of RALF family genes from different *F. vesca* tissues was depicted using heatmap3 package in R [42] from Transcripts Per Kilobase Million (TPM) values calculated by Li et al. [34, 43, 44, 45, 46].

## Infection of *Fragaria x ananassa* fruits

*F. × ananassa cv*. Alba plants were grown in the greenhouse at 25°C and 16 h of light. White (21 d after anthesis) and red fruits (28 d after anthesis) were harvested and infected. Each treatment contained at least three biological replicates. *Colletotrichum acutatum* (Isolate Maya-3, from CRIOF-UniBo fungi collection) and *Botrytis cinerea* strain *B05.10* were grown on PDA plates for 15 d. Detached fruits were dipped for 30 s in a $10^6$ conidia per mL suspension or in water as negative control and incubated in plastic bags for 24 h at approximately 21°C (room temperature) as reported in Guidarelli et *al*. (2011) [29].

## RNA extraction and qRT-PCR analysis

The surface of experimental fruits was excised with a scalpel and immediately frozen in liquid nitrogen. RNA was extracted according to Gambino et *al*. [47], run on an 2% agarose gel and quantified with NanoDrop™ 3300 for integrity and quality control, respectively. The cDNA was prepared from 1 µg of RNA using Promega ImProm-II™ Reverse Transcription system. Quantitative RT-PCR analysis was performed using ThermoFisher MAXIMA SYBR GREEN/ROX QPCR 2x supermix. The relative accumulation of RALF transcripts was calculated using standard curve method and *Elongation Factor 1* gene as reference (*XM_004307362.2)* [48]. Primers for RALF gene expression analysis were designed on *F. × ananassa*, *F. vesca* subgenome sequences and specificity was verified by observing a single peak in the dissociation curve for each primer pair. All primers used for gene expression analyses are listed in S2 Table.

## Statistical analyses

RALF and eGFP reporter transcripts were quantified as the average of three independent biological replicates, each formed by a group of at least three treated fruits. For eGFP quantification, all mean normalized expression values were expressed relatively to the negative empty pKGWFS7 vector infiltrated fruits. Student t-test was used to assess statistical significance between samples and controls.

## *In silico* prediction of regulatory elements in *FanRALF3-1* and *FanRALF6-1* promoters

Analysis of regulatory elements required for pathogen-induced transcripts was performed by examining pathogen-upregulated and downregulated transcripts found in datasets of red strawberry (*F. × ananassa)* fruits 24 h post infection with *C. acutatum* [29] and *B. cinerea* [49]. Promoters of 87 upregulated and 36 downregulated genes were analysed in response to *C. acutatum* infection, whereas 97 upregulated and 16 predicted promoters were analysed *B. cinerea* infected fruits. For each gene, 1500 bp upstream the ATG start codon were analyzed from the *F. vesca* genome v4.0.a1 assembly. MotifLab software v1.08 [50] was used for *in silico* analysis

using PLACE database for Motif Scanning [51], and AlignACE algorithm for Motif Discovery [52] and motif similarity analysis algorithm for motifs comparison. Briefly, Motif Scanning was performed using the Simple Scanner program with default parameters. A regulatory element was considered statistically significant to expression by performing the same Motif scanning program on randomly generated DNA sequences starting from input predicted promoter sequences using a third order background model. The frequency measured for each cis-acting element on random DNA was used as background occurrence for statistical significant evaluation using a binomial test with p-value threshold of 0.05. Motif Discovery was performed using AlignACE method with default parameters and motif significance were calculated as mentioned above for Motif scanning method.

### *Fragaria* × *ananassa* RALF3 promoter characterization

Preliminary assessment of putative promoter allelic variants were conducted on *F.* × *ananassa cv.* Alba and *cv.* Florida Elyana. Genomic DNA was extracted from leaves using Wizard$^{®}$ Genomic Purification kit (Promega). The Plant Tissue Protocol (Manufacturer protocol 3.E.) was modified adding two consecutive chloroform:isoamyl alcohol (24:1) purification steps after Protein Precipitation solution was added and before 2-propanol precipitation. The *FanRALF3-1* putative promoter was amplified using primers For 5′-TGCATCTGTTACATCAT CCCTTG-3′ and Rev 5′-GTAGTCGACTCTCCCATCTTG-3′, cloned into pGEM$^{®}$-T easy vector (Promega). Five clones for each variety were sequenced and aligned with the *F.* × *ananassa cv. Camarosa* genomic sequence available from GDR, using Clustal Omega.

### Cloning of progressively truncated promoters and *agrobacterium*-mediated transient transformation

The upstream sequence of *FanRALF3-1* was PCR-amplified starting from *F.* × *ananassa cv. Elyana* genomic DNA, using primer 5′-GGGGACCACTTTGTACAAGAAAGCTGGGTNCTGA AAGGACAAAAC ATTTTCT-3′ as the reverse primer for all promoter fragments, 5′-GG GGACAAGTTTGTACAAAAAAGC AGGCTNNTGCATCTGTTACATCATCCCTTG-3′ as the forward primer for the complete promoter fragment (T6), 5′-GGGGACAAGTTTGTACAAAA AAGCAGGCTNNTGCTTAAGTGGCTCTCAAAG-3′ as the forward primer for the 400 bp fragment (T4) and 5′- GGGGACAAGTTTGTACAAAAAAGCAGGCTNNCCGCTAAGTGGTTCAA TTCA-3′ as the forward primer for the 200 bp fragment (T2). Truncated *FveRALF3*-1 promoter constructs and a double p35S promoter used as a positive control were cloned into pDONR222 using the Gateway BP reaction and consequently cloned into pKGWFS7 (S1 Fig) vector by LR Reaction. The resulting plasmids were then introduced into chemically competent *Agrobacterium tumefaciens* strain EHA105 by heat shock transformation. Briefly, liquid nitrogen-frozen cells were thawed, and then incubated for 5 min at 37°C with 1 μg of plasmid DNA. The cells were then incubated at 30°C for 2 h in LB medium with shaking and plated. Positive colonies were then grown in selective media (Rifampicin 100 μg/mL and spectinomycin 50 μg/mL) until the culture reached an $OD_{600}$ of 0.8. Cells were collected by centrifugation and resuspended in fresh MS medium (Murashige and Skoog Basal Medium 4.4 g/L plus 20 g/L sucrose) and grown to $OD_{600}$ = 2.4. At the end, acetosyringone (4'-Hydroxy-3',5'-dimethoxyacetophenone) was added to the culture to final concentration of 200 μM. At least three white attached fruits for each condition were injected with about 3–5 mL of the *Agrobacterium* solution using a needle and syringe until culture filled strawberry fruit tissues. Five days after agroinfiltration, fruit were harvested and infected with *C. acutatum* conidial suspension or mock-inoculated with water, as was described above. After 24 h the fruits were dissected and

one half was used for RNA extraction and eGFP transcript analysis, and the other half was used for histochemical assay of GUS activity.

## Histochemical GUS assay

Surface tissue and longitudinal sections of infected and mock- infected fruits were cut with a razor blade and dipped in GUS staining solution (50 mM Na-phosphate (pH 7.5), 10 mM EDTA, 1 mM 5-bromo-4chloro-3-indolyl-glucuronide (X-gluc), 0.1% Triton X-100, 0.5 mM potassium ferricianide and 5% (w/v) polyvinylpyrrolidone-40 (PVP)). Strawberry tissues were incubated overnight at 37 ˚C, and then maintained at 4 ˚C in absolute ethanol until being photographed.

## 3D modelling of FanRALFs interaction with MRLK and LLG2 proteins

Homology models of *Fan*RALF3, in complex with FERONIA and LLG2 were generated using Modeller [53](v9.19) package using the complex of *A. thaliana* RALF23, LLG2 and FERONIA (PDB: 6a5e) as the template. ClustalX was used to create alignments of different components of the complex: *Fan*RALF3 (GDR: snap_masked-Fvb2-2-processed-gene-47.50-mRNA-1), *F. × ananassa* FERONIA MRLK47 (GDR: Uniprot: A0A1J0F5V4) and *F. × ananassa* LLG2 (GDR: maker-Fvb3-4-snap-gene-34.65-mRNA-1) with the *A. thaliana* proteins in the *A. thaliana* complex. PyMOL Molecular Graphic Systems (Schröedinger, LLC) was used for the analysis of the homology models and generation of the figures.

# Results and discussion

## Identification of RALF gene family members in *fragaria vesca*

RALF proteins belonging to 51 plant species have been previously classified in four clades based on sequence conservation [6]. Typical distinctive amino acid sequence motifs, such as the RRILA proteolytic cleavage site [7] and the YISY receptor binding site, are present in RALF peptides of clade I to III, and missing in clade IV, which contains more divergent members of the family. Nine RALF genes have been previously identified in the *Fragaria vesca* v1 genome.

In order to classify the members of RALF gene family in *Fragaria vesca*, the recent genome annotation (v4.0.a2) [34] was searched using 'RALF' as keyword gene name, revealing the presence of 13 RALF genes. These *F. vesca* RALF (*FveRALF*) [54] genes are named with progressive numbers according to their chromosome position, from Chr1 to 6 (Table 1), with one RALF gene in Chr1, two in Chr2, -3, and -5, and three in Chr4, and -6. No RALF genes are found in Chr7. Out of the nine RALF genes previously reported by Campbell and Turner [6], eight genes are confirmed both for identity and chromosome position. These are the gene08146 (corresponding to *FveRALF2*), *gene10567 (FveRALF3), gene02376 (FveRALF4)], gene02377 (FveRALF5), gene06579 (FveRALF6), gene06890 (FveRALF8), gene10483 (FveRALF9), gene22211 (FveRALF13)* (Table 1). The *gene00145*, previously annotated as gene encoding for a protein with the typical RALF motifs RRILA and YISY [6], was discarded since in the new v4.0.a2 annotation its sequence corresponds to gene *FvH4_6g07633* encodes for a shorter protein lacking most of the conserved motifs.

In order to find new, previously unannotated RALF genes, a BLASTn analysis was performed on the genome of the *F. vesca* (v4.0.a1 assembly) [55], using the 13 known *FveRALF* genes as query sequences. The BLAST search identified a putative new *FveRALF* gene located on chromosome 2 (Fvb2:15886205..15886408) and homolog to *FveRALF10* and coding for a protein lacking the RRILA cleavage site and the YISY active site, and having only two out of four cysteines at conserved positions, but presenting the RALF conserved domain in the C-terminal part of the protein sequence (S2 Fig). This hypothetical new RALF gene was named

**Table 1. List of RALF genes identified in *F. vesca*.**

| RALF genes | v4.0.a2 | Former annotation | Chromosome localization | clade | *Arabidopsis* ortholog |
|---|---|---|---|---|---|
| *FveRALF1* | FvH4_1g16140 | gene23829 | Fvb1:9232850..9233915 | III | *RALF34* |
| *FveRALF2* | FvH4_2g13590 | gene08146* | Fvb2:11869901..11870362 | III | *RALF4* |
| *FveRALF3* | FvH4_2g25351 | gene10567* | Fvb2:20606722..20607381 | I-II | *RALF33* |
| *FveRALF4* | FvH4_3g09010 | gene02376* | Fvb3:5266873..5268430 | III | *RALF19* |
| *FveRALF5* | FvH4_3g09020 | gene02377* | Fvb3:5269449..5270363 | III | *RALF19* |
| *FveRALF6* | FvH4_4g13190 | gene06579* | Fvb4:16794712..16795349 | IV | *RALF32* |
| *FveRALF7* | FvH4_4g13250 | gene06566 | Fvb4:16854377..16855101 | III | *RALF24* |
| *FveRALF8* | FvH4_4g18001 | gene06890* | Fvb4:21972673..21974193 | I-II | *RALF33* |
| *FveRALF9* | FvH4_5g21290 | gene10483* | Fvb5:12816493..12816858 | IV | *RALF33* |
| *FveRALF10* | FvH4_5g26840 | gene41610 | Fvb5:18214922..18215453 | IV | *RALF25* |
| *FveRALF11* | FvH4_6g36290 | gene42389 | Fvb6:28540907..28541125 | IV | *RALF32* |
| *FveRALF12* | FvH4_6g41850 | gene42489 | Fvb6:32780751..32781347 | IV | *RALF5* |
| *FveRALF13* | FvH4_6g06520 | gene22211* | Fvb6:3818620..3820001 | I-II | *RALF33* |
| *FveRALF14* | - | - | Fvb2:15886205..15886408 | IV | *RALF25* |
| *FveRALF15* | | FvH4_3g15010 | Fvb3:9301819..9303781 | III | *RALF19* |

V4.0.a2 annotation according to Li et al.,[35]; Former annotation corresponds to v1.1 (geneN˚) or v4.0.a1 (FvH4_N˚) [51]. Asterisks (*) indicate those genes that were previously identified in v1 *F. vesca* genome [6] and confirmed in the v4.0.a2 annotation version; clade classification according to Campell and Turner [6].

*FveRALF14* (Table 1). Furthermore the BLASTn analysis revealed the presence of another predicted new *FveRALF* gene, with two RALF conserved domains both homolog to *Arabidopsis AtRALF19* and alternatively translated in frame 1 or frame 2. This gene was annotated also in the previous V4.0.a1 version of *F. vesca* genome (*FvH4_3g15010*) and here named *FveRALF15* (Table 1).

The FveRALF protein members were aligned using Clustal Omega (S3 Fig) and classified in clades according to Campbell and Turner [6] (Table 1). Overall, FveRALF clustered in three major clades (Table 1) instead of four, since FveRALF proteins of clade I and II were not distinguishable: namely clade I-II (FveRALF3, FveRALF7 FveRALF8, FveRALF13), clade III (FveRALF1, FveRALF2, FveRALF4, FveRALF5, FveRALF9 and FveRALF15), clade IV (FveRALF6, FveRALF9, FveRALF10, FveRALF11, FveRALF12, FveRALF14). The FveRALF include two members that aligned well with *Arabidopsis AtRALF32* (FveRALF11, FveRALF6), two with similarity to AtRALF25 (FveRALF10, FveRALF14), three closely matching AtRALF19 (FveRALF4, FveRALF5, FveRALF15), four similar to AtRALF33 (FveRALF3, FveRALF8, FveRALF13, FveRALF9), and one respectively to AtRALF4 (FveRALF2), AtRALF5 (FveRALF12), AtRALF24 (FveRALF7) and AtRALF34 (FveRALF1). All *FveRALF* genes are predicted to contain a single exon and no introns, except for predicted *FveRALF15* which possesses two exons and one large predicted intron. Interestingly, the transcript *FveRALF3* originates from a putative Natural Antisense Transcript (NAT) generating region and its complementary sequence encodes for the 3' untranslated region of a heat shock factor binding protein gene (*FvH4_2g25350*).

## Identification, evolution and chromosome organization of RALF genes in *F. x ananassa*

*F. vesca* RALF gene sequences were used as query against a database of predicted proteins (v1.0.a1 Proteins source [35]) in *F. × ananassa cv. Camarosa* (Fxa). Fifty RALF members were identified in the *Fxa* octoploid strawberry (S1 Table). Paralogous genes (orthologs to particular

*FveRALF* genes) were identified in the *Fxa* subgenomes from the alignment and phylogenetic analysis (S4 Fig and Fig 1A). Progenitors lineage was inferred from chromosome localization (Fig 1B) according to the prediction by Edger *et al.* [35]. Fourteen genes out of these 50 are localized in the *F. vesca* subgenome, 15 members in the *F. nipponica* subgenome, 13 in the *F. iinumae* subgenome and only eight in the *F. viridis* subgenome (Fig 1C). It is important to note that these assignments were based on the best subgenome match in the 'Camarosa' genotype. This is one accession and octoploid strawberry subgenomes structure may not be the same among all genotypes at this ploidy.

*Fxa RALF* genes *(FanRALF)* were named based on corresponding *FveRALF* orthologs and the subgenome localization, with progressive numbers from 1 to 4 to indicate *F. vesca*, *F. iinumae*, *F. nipponica* and *F. viridis* progenitors, respectively (accordingly to Edger *et al.*[35]), and progressive letters to nominate genes orthologous to the same *FveRALF* genes and localized on the same chromosome (e.g. *FanRALF8-2a* is one of the two *Fxa* genes orthologous to *FveRALF8* mapping on *F. iinumae* subgenome).

Gene homology and chromosome localization analysis showed that only four out of 13 *FveRALF* genes, namely *FveRALF1*, *FveRALF4*, *FveRALF7*, *FveRALF9*, have orthologs in all the four subgenomes. For all the other cases the *FveRALF* orthologs are not represented in all the

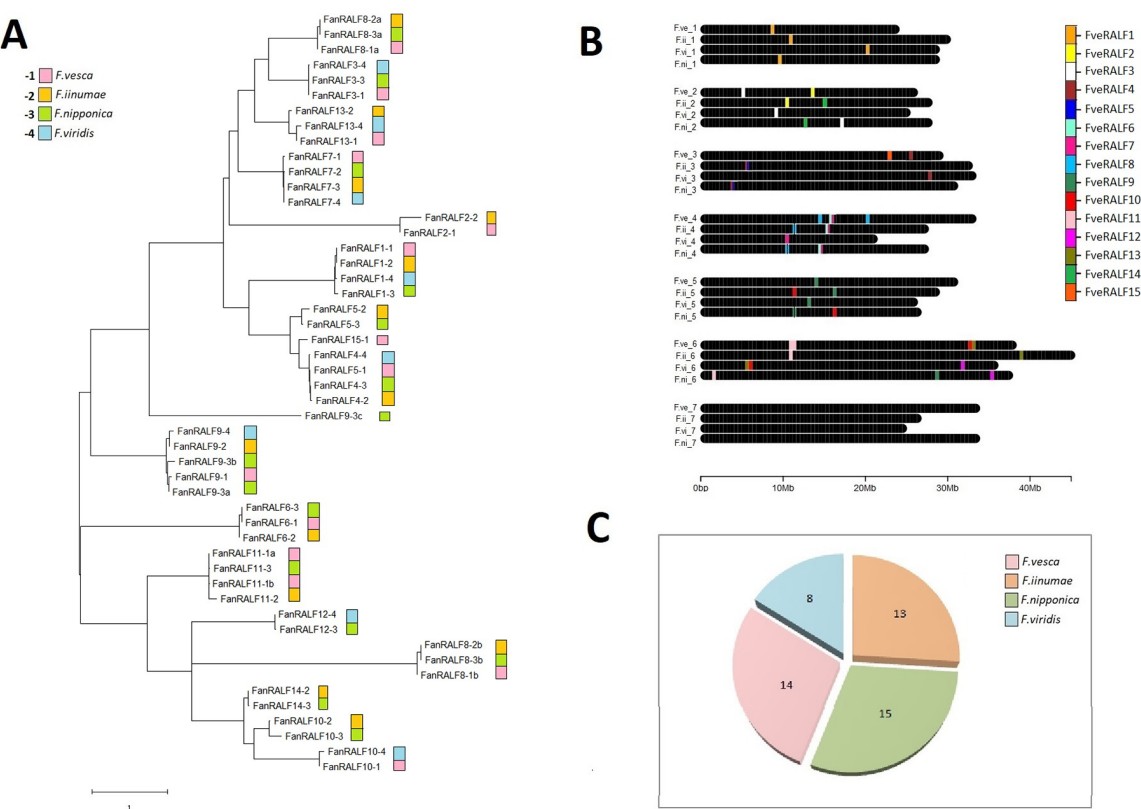

**Fig 1.** *Fragaria x ananassa (Fxa)* **RALF genes phylogentic analysis, evolution and chromosome organization.** (a) Phylogenetic tree was built aligning 50 *Fan*RALF protein sequences using MUSCLE. The tree is drawn to scale, with branch lengths measured in the number of substitutions per amino acidic site in the protein sequences. Gene annotations refer to *F. × ananassa cv. Camarosa* v1.0.a1 and are listed in S1 Table. Progenitor lineage were inferred from gene chromosome location (b), and pink was used for *F. vesca* subgenome, green for *F. nipponica*, light blue for *F. viridis* and orange for *F. iinumae*. The color legend also reports the code number used to name different *FanRALF* genes according to subgenome lineage, as is reported in S1 Table. (b) *FanRALF* genes chromosome spatial organization in the octoploid genome. (c) Pie chart showing total *FanRALF* gene members present in the four subgenomes.

different *Fxa* subgenomes, probably due to gene loss events occurred during evolution or poly-ploidy adjustment (Fig 1A). In particular, the *F. viridis* derived subgenome has the lowest number of RALF gene members and is lacking genes orthologous to *FveRALF2*, *FveRALF5*, *FveRALF6*, *FveRALF8*, *FveRALF10*, *FveRALF11* and *FveRALF14*. Similarly to *F. vesca*, no RALF genes are localized in Chr7 of the different progenitors. It is likely that in the *Fxa* genome some of the RALF genes are the result of duplication events. For example *Fan-RALF11-1a* and *FanRALF11-1b* are both orthologous to *FveRALF11* and are positioned close together on Chr6 (*F. vesca* subgenomes Fig 1B). Another atypical gene organization is found for *FanRALF15* and *FanRALF4-1* genes, located on Chr3 (*F. vesca* subgenome), which are annotated as single genes but contain two tandem RALF conserved domains, suggesting that a duplication event occurred during genome evolution. In addition, the *FanRALF7-4* gene is predicted to encode a 325 aa protein containing a conserved RALF domain within the first 104 amino acids and a domain homologous to chloroplastic NADPH-dependent aldehyde-reductase like protein, from aa 124 to 325. There are many potential explanations for how this novel domain architecture could occur, and it will be interesting to test for functional relevance of this particular variant.

Genes *FanRALF9-3a* and *FanRALF9-3b*, occur as NAT element on Chr 5 (*F. nipponica* subgenome), as was observed in *F.vesca* for *FveRALF3*.

The *F. viridis* derived subgenome contains the fewest detected putative RALF genes. This was also the case for R gene family in *Fxa* [35]. However, in contrast to the R gene family, there is not a clear dominance of *F.vesca* progenitor in the *FanRALF* gene family composition, since genes are similarly distributed in the *F. iinumae*, *F. nipponica* and *F.vesca* subgenomes (Fig 1C). As was speculated by Edger *et al.* [35] the lack of RALF genes in *F. viridis* subgenome could be related to the higher TE content of this subgenome which can cause both higher mutation rates and gene loss. *Fxa* RALF gene classification in specific clades agrees with the structure in diploid woodland strawberry, with 21 genes in clade IV, 17 genes in clade III, 12 in clade I-II (S1 Table).

## Transcriptome dataset analysis of RALF genes in *F.vesca*

To provide insights into the RALF gene members functions in strawberry, RNA-seq datasets that have been mapped onto the new genome annotation v4.0.a2 by Li *et al.* [34] were analyzed in different tissues and developmental stages. RALF members were grouped based on similar expression profile and hierarchical clustering resulted in four major RALF expression groups (Fig 2): i) RALF genes specifically expressed in mature male gamete *(FveRALF4*, *FveRALF5*, *FveRALF10*, *FveRALF11)*; ii) a gene expressed only in two anther developmental stages (*FveR-ALF2)* iii) *FveRALF3* and *FveRALF12* genes mainly expressed in roots, and in roots after two days after infection with *Phytophthora cactorum* iv) and genes mainly expressed in different fruit developmental stages (*FveRALF1*, *FveRALF6*, *FveRALF7*, *FveRALF8*, *FveRALF9* and *FveRALF13)*.

Contrary to what has been observed in *Arabidopsis*, where clade IV RALF genes were highly expressed in flower tissues [6], the woodland strawberry *FveRALF* transcripts included in each of the four expression groups belong to different clades, suggesting that members of the same clade are have different roles in different contexts. The highly specific expression of four *FveR-ALF* genes in male gamete and late stage of anther development (*FveRALF4*, *FveRALF5*, *FveR-ALF10*, *FveRALF1)* shown by the heatmap (Fig 2), suggests a role for these RALF genes in the ovule-pollen, cell-cell communication during the sequence of events precisely regulated during fertilization. A recent study reports that in Arabidopsis *AtRALF34* gene, expressed in the ovule, competes with *RALF4* and *RALF19*, expressed in the pollen tube, for binding to BUPs

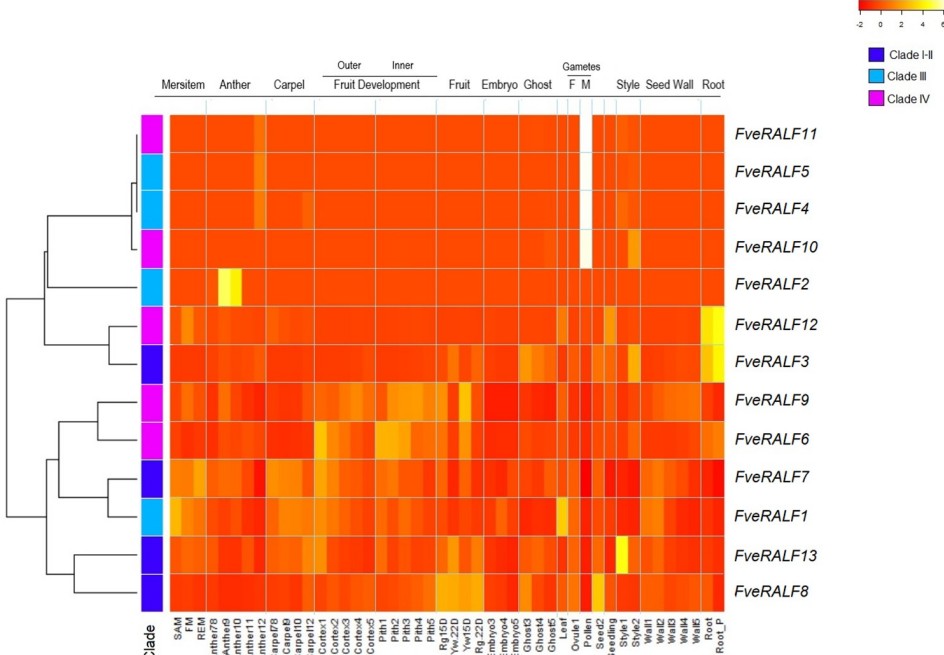

**Fig 2. Detection of RALF transcripts in *F. vesca*.** The heatmap reflects the expression profile (as Transcription per Kilobase Million (TPM)) of *FveRALF* members (rows) in different tissues and developmental stages (column) of *F. vesca*. Labels at the bottom specify the tissues and the stages, whereas labels at the top group tissues and stages for organs. Dendrogram on the left shows rows relationship according to similar expression profile. RALF members classification in clades are shown: in green clade I, in blue clade II, in light blue clade III and magenta clade IV. SAM (shoot apical meristem), FM (floral meristem), REM (receptacle meristem), Anther7-8 (identified by stomium development and appearance of a preliminary lobed structure), Anther9 (microspore mother cells start meiosis), Anther10 (microspores are loose in the locule after callose wall holding tetrads disaggregates), Anther11 (pollen mitotic division occurs), anther12 (no visible change in anther development), Carpel7/8 (round carpel primordial reach the receptacle apex), Carpel9 (bowling pin shaped carpel primordial), Carpel10 (carpel is dived in almost equal apical and basal part by a central constriction), Carpel11 (style is elongated and became twice in length than the ovary base), Carpel12 (carpels have music note shape and styles are separated from each other), Cortex1 and Pith1 (flower just opened), Cortex2 and Pith2 (at about 3 DPA, when pollination occurs), Cortex3 and Pith3 (at about 6 DPA), Cortex4 and Pith4 (at about 9 DPA), Cortex5 and Pith5 (at about 12 DPA), Rg15D and Rg22D (Red Ruegen receptacle tissue at 15 DPA and at 22 DPA), Yw15D and Yw22D (Yellow Wonder receptacle tissue at 15 DPA and at 22 DPA), Embryo3 and Ghost3 (embryo and seed without embryo inside at about 6 DPA characterized by heart shape), Embryo4 and Ghost4 (at about 9DPA, with immature cotyledons), Embryo5 and Ghost5 (ad about 12 DPA, mature embryo which fill up entire ovules), Leaf (young trifoliate leaves), Ovule1 and Pollen (collected from just open flower), Seed2 (complete achene from mature fruit), Seedling (complete seedling at 10 days post germination), Style1 (style and stigma from just open flowers), Style2 (style from flower at about 3 DPA), Wall1 (carpel wall from just open flower), Wall2 (carpel wall at about 3 DPA), Wall3 (carpel wall at about 6 DPA), Wall4 (carpel wall at about 9 DPA), Wall5 (carpel wall at about 12 DPA), Root (collected from 7 week old plants grown in aerated hydroponic culture) and Root_P (after 2 days of inoculation with *Phytophthora cactorum*.

and ANXs receptors [13]. The interaction between *AtRALF34*, expressed by the female gamete, and receptor complex formed by BUPS1/2 and ANX1/2, present on pollen tube membrane leads to pollen tube rupture and sperm release [13]. The competitive binding of *At*RALF4 and *At*RALF19 to this receptor complex suggest that they have a redundant function in regulating pollen tube growth and integrity [16]. It is possible that these RALF genes functional redundancy is conserved in woodland strawberry.

Among the *FveRALF* transcripts expressed in flower and fruit organs, *FveRALF1* and *FveRALF7* are the most abundant at the early stage of development in shoot apical meristem (SAM), floral meristem (FM) and receptacle meristem (RM), with *FveRALF1* also being the

family member most highly expressed in the leaves. As for fruit, *FveRALF8* is the gene most highly expressed in mature fruits, whereas *FveRALF6*, *FveRALF7*, *FveRALF9* are detected during fruit growth and in the mature organ at 15 d post anthesis (15 DPA) both in Yellow Yonder and Red Rugen genotypes (the two *F. vesca* genotypes used for RNA seq) and *FveRALF3* and *FveRALF13* transcripts are most abundant in the mature fruits at 20 DPA. *FveRALF1*, *FveR-ALF6*, *FveRALF7* and *FveRALF13* expression decreases during fruit development both in the inner and the outer tissues of fruit while *FveRALF9* expression gradually increases with fruit development.

In a recent work, Jia et al. [56] analyzed the expression of the woodland strawberry (*F. vesca*) Malectin Receptor Like Kinases (MRLK) also known as the *Catharanthus roseus* RLK-like proteins (RLK1Ls). *F. vesca* MRLKs are encoded by more than 60 genes, and more than 50% of these are expressed during fruit development. The majority of fruit *FvMRLK* genes are expressed at high level only at the early stage of fruit ripening, and decrease at ripe stages. Transiently silencing and overexpression of *MRLK47* in strawberry fruit, severely affects ripening regulation [50]. Moreover, a recent report describes how MRKL47 changes the sensitivity of ripening-related genes to ABA, a key hormone for strawberry fruit ripening [57]. Consistently, both RALF and ABA were found to be FERONIA-mediated cross-talk signals in stress-response and cell growth in *Arabidopsis* [58]. FERONIA receptor is known to be important for cell-wall integrity and in $Ca^{2+}$ signaling [59], both known to be important during fruit growth and ripening [60]. The similar expression profile of *Fve*MRLKs to the one we observe here for *FveRALF1*, *FveRALF6*, *FveRALF7* and *FveRALF13* supports the idea that the RALF-MRLK signalling plays an important role for also for strawberry fruit development. Future studies will test the interaction between RALF peptides and *Fve*MRLK *in vivo*.

## Expression profile of RALF genes in *Fragaria x ananassa* fruit and induction upon pathogen infection

RALF peptides are known to play a role in plant-pathogen interaction [24] since they were found to negatively regulate plant immunity response in *Arabidopsis* [12]. They were also found to be secreted by fungal pathogen as crucial virulence factors [25, 61]. Furthermore, it was reported that genes homologous to *AtRALF33* were upregulated both in tomato (*Solanum lycopersicon)* and commercial strawberry (*F. × ananassa)* susceptible ripe fruits interacting with *C. gloeosporioides* and *C. acutatum*, respectively [29, 31, 32].

For this reason, the transcript levels of *FanRALF* genes were assessed in *F. × ananassa* fruit at two different ripening stages and upon infection with two different fungal pathogens, *C. acutatum* or *B. cinerea*. Fruit *FanRALF* gene targets were chosen based on the *FveRALF* gene homologs expressed in fruit (Fig 2) since the *F. vesca* progenitor is reported to have the most dominant transcripts detected among the different subgenomes [35]. Therefore, primers were designed to amplify genes that are expressed in fruit (*FanRALF1-1*, *FanRALF3-1*, *FanRALF6-1*, *FanRALF7-1*, *FanRALF8-1*, *FanRALF9-1 and FanRALF13-1*), which included orthologs to *AtRALF33* (*FanRALF3-1*, *FanRALF8-1*, *FanRALF9-1* and *FanRALF13-1)*.

*FanRALF3* transcripts show a significant increase in abundance upon infection with either pathogen in the susceptible ripe stage, whereas *FanRALF9* expression decreases in white fruits upon *C. acutatum* but not upon *B. cinerea* infections (Fig 3). The expression of *FanRALF8* and *FanRALF13* were not affected by infection, neither in white nor in red fruits (Fig 3). Out of the other *FanRALF* genes analyzed, only *FanRALF6* shows a clear downregulation in infected fruits at both ripening stage and in response to both pathogens, while *FanRALF1 and FanRALF7* gene expression is significantly decreased only in red ripe stage with *B. cinerea* and in white stage with *C. acutatum*, respectively.

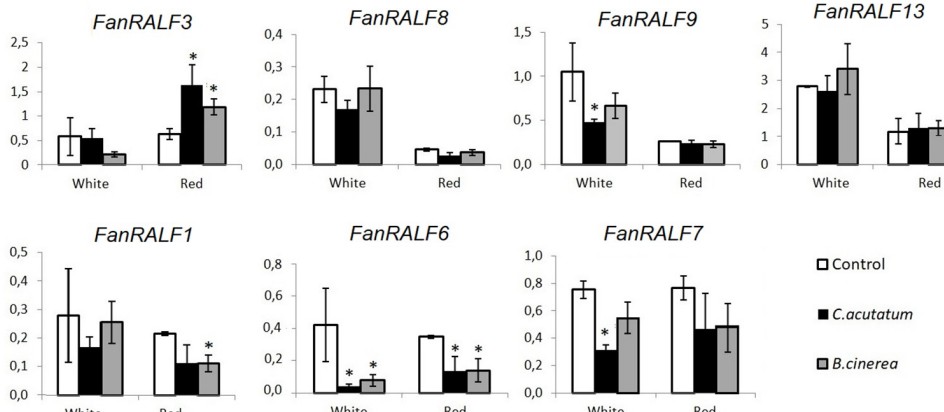

**Fig 3. Expression of different RALF transcripts upon infection with pathogens.** qRT-PCR analysis of different RALF genes (at the top the RALF-like 33 genes *FanRALF3*, *FanRALF9*, *FanRALF8* and *FanRALF13*) in Fxa strawberry fruits at different ripening stages (white and red) 24 h post infection with *C. acutatum* and *B. cinerea*. Histogram bars represent three biological replicates average relative expression and black lines represents standard deviations. Student's T test was used to calculate statistical significance between infected and control samples p< 0.05 (*).

The expression profiles of fruit RALF genes on *Fxa* strawberry fruits infected with two post-harvest pathogens are consistent with our previous results [30]. This report showed that transient overexpression of a *FanRALF33-like* gene (here now named *FanRALF3*) affected disease susceptibility. Notably, the *FanRALF3* and *FanRALF13* genes encode mature peptides differing only by two amino acids (S2 Fig) but accumulate differently to pathogen infection, suggesting that the genes possess subfunctionalized promoters that allow responses to different stimuli, or that the two amino acid difference leads to contrasting roles for the two different peptides.

Furthermore, N-terminal sequence alignment of *Fan*RALF1-1, *Fan*RALF3-1, *Fan*RALF7-1, *Fan*RALF8-1a and *Fan*RALF13 peptides with *At*RALF23 (Fig 4A) shows that residues directly involved in *At*RALF23 binding with *At*LLG2 are conserved, suggesting that *Fan*RALF peptides in strawberry fruits may be the strawberry orthologs in the LLG-MRLK heterotypic complex. Consistently, the homology models of *Fan*RALF3-1 peptide interaction with *Fan*MRLK47—

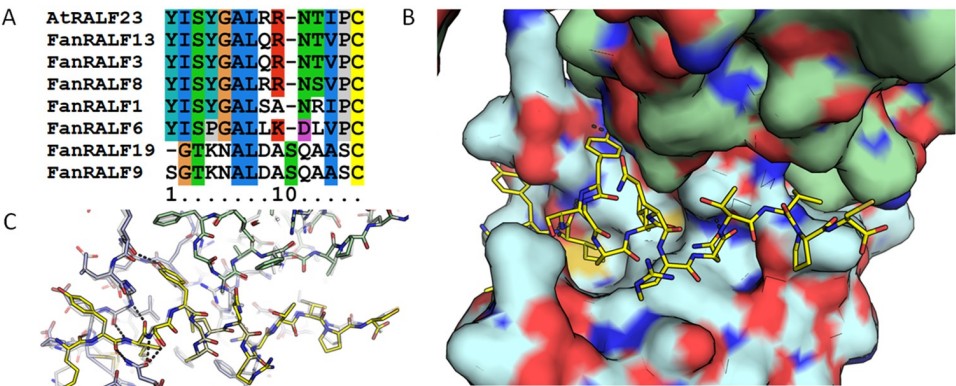

**Fig 4. Model of *Fan*RALF3, *Fan*MRLK47 and LLG2.** (A) multiple sequence alignment of *Fan*RALFs and *At*RALF23. (B) Model of the ternary complex of *Fan*RALF3, *Fan*MRLK47 and LLG2 based on *At* complex (PDB:6a5e). C. overlay of the *Fan* complex model and the *At* complex, showing high conservation of both the RALF and the two receptors.

the MRLK mostly expressed in *Fxa* fruits [56] and with the *Fxa* LLG2 homolog (maker-Fvb3-4-snap-gene-34.65), show that the structural components necessary to bind MRLK47 and LLG2 are conserved in *Fan*RALF3 (Fig 4B and 4C), suggesting a similar binding mechanism and complex formation. In *Arabidopsis*, *At*RALF23 binding to FERONIA and LLG proteins leads to negative immunity response regulation, and this hypothesis can be tested by analysis of *Fan*RALF3-1 binding to MRLK47 in fruits.

Finally, the expression of *FanRALF1*, *FanRALF6* and *FanRALF9* decreases during infection, thus it is possible that these *FanRALF* gene members may play different roles than *RALF3* homologs during plant-pathogen interaction and immune response.

## *FanRALF3* promoter analysis in *Fragaria x ananassa* subgenomes and varieties

As shown above and reported in previous studies, RALF gene expression is triggered by different biotic and abiotic stimuli, however the signaling events regulating its expression are not yet known. Identification of the promoter elements necessary for gene induction by fungal pathogens could provide a basis to identify the transcription factors and other mechanisms involved in immunity signaling and ultimately provide the necessary knowledge to develop synthetic pathogen-responsive promoters to fight infections. Among *FanRALF* family genes, *FanRALF3* shows increases in transcripts in response to pathogen, and its overexpression in strawberry fruits is related to susceptibility [30]. For this reason *FanRALF3* was chosen for promoter truncation analysis.

To study *FanRALF3* putative promoter function in *F.* × *ananassa*, the upstream sequence conservation among the *Fxa* subgenomes was assessed. *FanRALF3-1* sequence from the 3'UTR of upstream flanking gene (annotated as '*maker-Fvb2-2-augustus-gene-47.69-mRNA-1*') and its ATG (590 bp) was used as input for BLASTn analysis against *Fragaria x ananassa cv.* *Camarosa* v1.0.a1. Five sequences were retrieved on *F. iinumae* Chr2-4, *F. nipponica* Chr2-1 and *F. viridis* Chr 2–3 (S4 Fig), indicating that the putative *FanRALF3* homoeologous promoter sequences in the the *Fxa* subgenomes are highly similar, except for *F. viridis*, already reported to be the most divergent and silent in octoploid genome [35].

To study allelic variability, the *FanRALF3-1* putative promoter sequence similarity was assessed also in genomes of *F.* × *ananassa* varieties with different susceptibility to fungal pathogens, the *cv. Florida Elyana* from Florida (U.S.A.) which is resistant to anthracnose disease [62], and *cv. Alba*, an italian variety which is highly susceptible to *C. acutatum* infection (https://plantgest.imagelinenetwork.com/it/varieta/frutticole/fragola/alba/59). The promoters were amplified with specific primers and five clones for each variety were sequenced and aligned with that of the v1.0.a1 genome sequence *cv. Camarosa* (S5 Fig). Only a single nucleotide polymorphism was detected between *cvs. Alba* and Florida Elyana. This suggests that the function associated with the *FanRALF*3-1 5' upstream sequence in octoploid strawberry may be based on small differences in sequence, or that factors upstream of the promoter are the basis of the mechanism of resistance.

## Prediction of *FveRALF3* promoter pathogen-responsive regulatory elements

The *FanRALF3-1* (from *F.vesca* subgenome) putative promoter sequence was chosen for pathogen-responsive regulatory element analysis because of the reported *F. vesca* subgenome dominance in octoploid genome [35]. Since *Fa-* and *FveRALF3* upstream putative regulatory sequences share 99% level of identity and the available *Fxa* pathogen-responsive transcriptome data have all been mapped onto *F. vesca* genome, the analysis of the *FanRALF3-1* promoter

regulatory elements responsive to pathogen infection were carried out on *F. vesca* genome *FveRALF3* promoter.

The 656 bp genomic sequence upstream of *FveRALF3* ATG start codon was analysed using truncation analysis. *FveRALF3* putative promoter sequence was compared with known transcription factor binding sites of genes known to be regulated in *Fxa* strawberry fruit upon *C. acutatum* and *B. cinerea* infections [29, 49] in the PLACE database [51] (Motif Scanning analysis). These latter sequences and *FveRALF3* putative promoter were then scored for motif frequency (Motif Discovery analysis) (Fig 5).

Motif scanning analysis of cis-acting elements enriched for fungal-induced transcripts revealed the presence of an element initially identified as initiator of PsaDb gene promoter (INRNTPSADB, PLACE ID: S000395). The putative promoter lacks a TATA-box, which is common to *FveRALF3* and the majority of the 5' upstream sequence (78 out of 87 (90%)) of the *C. acutatum* responsive genes promoter sequences and 92 out of 97 (94%) of *B. cinerea* genes (Fig 5). Other TATA-like elements such as TATABOX2 (PLACE ID: S000109) and TATAPVTRNALEU (S000340), which have the role of recognition and initiator of transcription complex [63, 64] were also found in several putative promoter sequences (59 (67%) from *C. acutatum* upregulated genes and in 41 (42%) of *B. cinerea* induced genes). Among the group of genes upregulated by *C. acutatum*, 33 (37%) and 47 (57%) predicted promoters

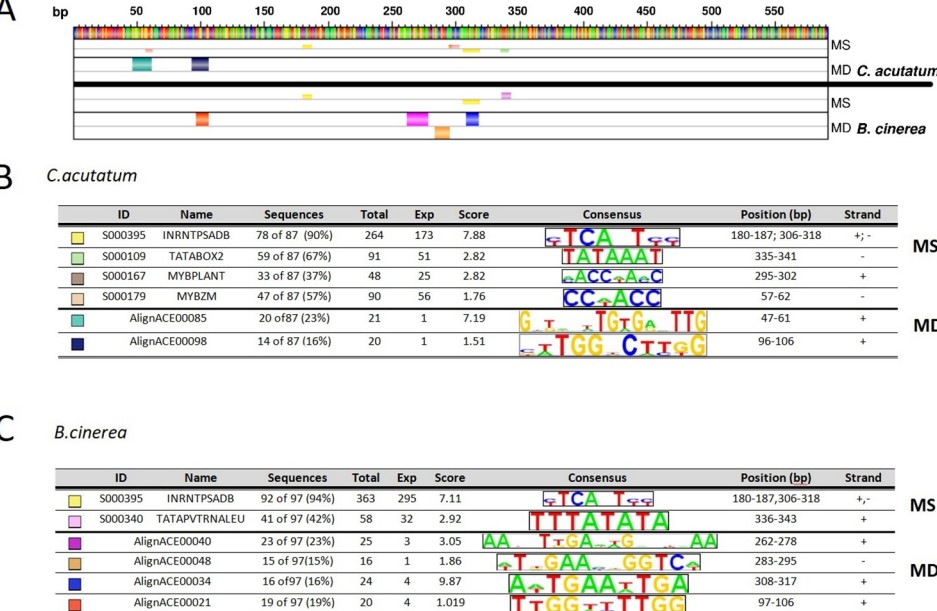

**Fig 5.** ***In silico* analysis of predicted regulatory elements in pathogen-induced *fragaria* genes.** (A) From the top: FveRALF3 gene putative promoter, Motif Scanning (MS) and Motif Discovery (MD) outputs resulting from the analyses of *C. acutatum*-induced genes, and MS and MD outputs from the analysis of *B. cinerea*-induced genes 24 h post-infection. Small colored boxes in MS indicate known regulatory elements from PLACE database found to be significantly abundant in the group of sequences analyzed and in *FveRALF3* putative promoter, while large coloured boxes in MD represent sequences found significantly enriched in the upstream sequences of genes analyzed. (B) and (C) are the color key respectively for *C. acutatum* and *B. cinerea* reported in (A). Codes and names of regulatory elements are reported, together with elements percentage abundance among sequences analyzed (Sequences), total count of elements found (total), element background frequency calculated as number of elements found in a third order random generated sequences (Exp), ranking score values calculated by MotifLab software using a binomial test with p-value threshold of 0.05 (Score). Consensus sequence of each element is shown along the position of the element in the putative *FveRALF3* promoter (Position bp) and the strand in which it is found.

possess MYBPLANT (S000167) [65] and MYBZM (S000179) [66] regulatory elements, which are binding sites for MYB-type transcription factors. Additional analysis on the putative promoters of Arabidopsis *RALF33* (*AT4G15800*] and its homologous in tomato (*S. lycopersicum Solyc09g074890.1)*, revealed the presence of the same binding site (data not shown). MYB proteins are a large family of transcription factors involved in regulation of many processes in plants, such as phenylpropanoid metabolism [67], ABA and JA signaling [68], responses to abiotic and biotic stress [69], cell death and output from the circadian clock. MYB proteins generally interact with basic helix-loop helix (bHLH) family member and are regulated by cytosolic WD40 repeat proteins through formation of MYB/bHLH/WD40 dynamic complexes, which regulate various gene expression pathways [70]. Interestingly, MYB46 is involved in enhancing *B. cinerea* resistance through down-regulation of cell wall associated genes (CESA) during early stage infection [71]. *Arabidopsis* T-DNA insertion mutants of genes regulated by MYB46, such as the a zinc-finger containing protein *gene zfp2*, the Basic Helix-Loop-Helix TF *bhlh99*, the AUX/IAA-type transcriptional repressor *pap2* and the *At1g66810 gene* coding for a Zing Finger Transcription Factor, showed enhanced susceptibility to the necrotrophic fungal pathogens *B. cinerea* and *P. cucumerina* [33]. The transcript analysis of these four mutants revealed a coincident upregulation of *RALF23*, *RALF24*, *RALF32* and *RALF33* [33] showing that MYB46 and RALF genes are part of a co-expression network, and possibly functionally related.

A Motif Discovery analysis was performed using AlignACE algorithm [52] on the putative promoter sequences of both *Fve*RALF3 and the identified *C. acutatum* and *B. cinerea* strawberry upregulated genes, and significantly overrepresented motifs were assessed. The motifs identified using MotifLab software are named 'AlignACE' followed by progressive numbers. For *C. acutatum* genes group the AlignACE00085 element (consensus GxTxxxTGTGAxTTG) was found in 20 (23%) predicted promoters, and is partially overlapping with MYBZM elements at the position between bases 57 and 62 of *FveRALF3* putative promoter. The AlignACE00098 (xxTGGxCTTGG) element was found in the 14 (16%) *C. acutatum* upregulated genes and align to the elements AlignACE00021 (TTGGxxTTGG) found in *B. cinerea* upregulated group (19% of sequences analyzed). It is tempting to speculate that the position of this element might be a regulatory component important for *FveRALF3* fungal-induced expression. Furthermore in *B. cinerea* gene group the AlignACE00040 (AAxxTTGAxxGxxxAA), AlignACE00048 (TxGAAxxGGTC) and AlignACE00034 (AxTGAAxTGA) motifs, located between bases 262 and 317 in *FveRALF3* putative promoter, were found significantly overrepresented.

Besides upregulation of *FanRALF3*, both pathogen infections led to the downregulation of *FanRALF6* transcript accumulation (Fig 3), suggesting a role also for this gene during fruit defence response. In order to explore why these two genes showed opposite behaviour and to highlight the presence of similar cis-regulatory sequences in *FanRALF3* and *FanRALF6* promoters, the predicted promoter sequence of *FanRALF6* was also scored for the precence of annotated cis-acting elements and motif discovery by analysing sets of downregulated genes in datasets from *C. acutatum* and *B. cinerea* infected fruit tissues (Fig 6A).

Similar to the *FanRALF3-1* predicted promoter, the *FanRALF6-1* upstream sequence (*F. vesca* progenitor) has very high identity (99%) with *F. vesca* (data not shown), thus the regulatory element analysis was performed considering 1500 bp from ATG of *FveRALF6* gene. Contrary to the high numbers of genes found upregulated upon *C. acutatum* and *B. cinerea* infections, examined in the *FveRALF3* promoter analysis, the number of genes downregulated similarly to *FveRALF6* is limited to 36 for *C. acutatum* and 16 for *B. cinerea*. The motif scanning analysis revealed that 100% of downregulated genes during *C. acutatum* early infection (Fig 6B) contain the INRNTPSADB element (S000395, [63]), found also in upregulated genes analysis (Fig 5), and known to particiapte in transcription initiation. Analysis also uncovered

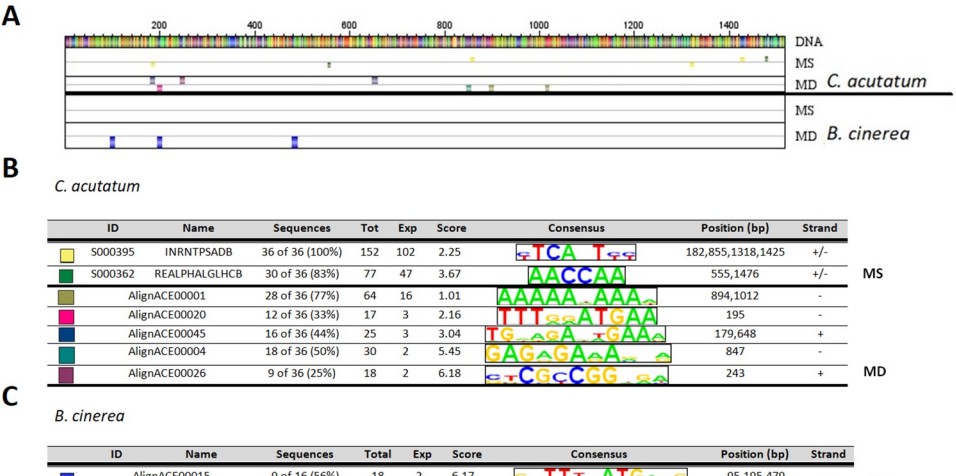

**Fig 6. *FanRALF6 In silico* analysis of predicted regulatory elements in pathogen-induced *fragaria* genes.** (A) From the top: the *FanRALF6* gene putative promoter, Motif Scanning (MS) and Motif Discovery (MD) outputs resulting from the analyses of *C. acutatum*-induced genes, and MS and MD outputs from the analysis of *B. cinerea*-induced genes 24 h post-infection. Small colored boxes in MS indicate known regulatory elements from PLACE database found to be significantly abundant in the group of sequences analyzed and in *FanRALF6* putative promoter, while large coloured boxes in MD represent sequences found significantly enriched in the upstream sequences of genes analyzed. (B) and (C) are the color key respectively for *C. acutatum* and *B. cinerea* reported in (A). Codes and names of regulatory elements are reported, together with elements percentage abundance among sequences analyzed (Sequences), total count of elements found (total), element background frequency calculated as number of elements found in a third order random generated sequences (Exp), ranking score values calculated by MotifLab software using a binomial test with p-value threshold of 0.05 (Score). Consensus sequence of each element is shown along the position of the element in the putative *FanRALF6* promoter (Position bp) and the strand in which it is found.

the "REalpha" element (REALPHALGLHCB, S000362), a feature initially identified in *Lhcb21* gene promoter of duckweed (*Lemna gibba)* [72, 73], that is involved in phytochrome regulation. This element is present in 30 out of 36 (83%) sequences analyzed with the consensus sequence AACCAA, and shares high similarity with sequences recognized by MYB family TFs. The motif has been identified in promoters of stress responsive genes, such as MYB1AT (S000408, aAAC-CAA) and MYBATRD22 (S000175, CTAACCA) (S3 Table). MYB binding motifs (MYBPLANT and MYBZM) were also found in *FanRALF3* and other genes upregulated upon fungal pathogen infection (Fig 5B). These preliminary analyses suggest a role of these conserved sequences in pathogen responsive gene regulation. On the other hand, the motif scanning analysis of *B. cinerea* downregulated genes did not show any significant result (Fig 6C).

In addition, the motif discovery analysis revealed the occurrence of conserved motifs among *C. acutatum* downregulated genes (Fig 6B) and in particular AlignACE00001 (AAAAAxAAAx) motif was found in 28 (77%), AlignACE00020 (TTTggATGAA) in 12 (33%), AlignACE00045 (TGxagAxtGAAa) in 16 (44%), AlignACE0004 (GAGAGAaAxxa) in 18 (50%) and AlignACE00026 (ctCGCGGxga) in 9 (25%) out of 36 sequences analyzed. The AlignACE00001 consensus motif (AAAAAxAAAx) is a A-reach sequence that has been reported to correlate with scaffold attachment regions [73] and in chromatin loop domain organization and transcription [74]. On the other hand, the AlignACE00004 motif (GAGAGAaAxxa) shows the typical sequences recognized by GAGA-binding proteins involved in chromatin silencing modification [75]. Motif discovery analysis performed on *B. cinerea* induced-downregulated genes revealed the enrichment of AlignACE00015 motif (gxTTTxATGaxg) which was found in 9 out of 16 (56%) sequences analyzed.

### *FveRALF3* promoter-reporter assay

In order to explore the functional relevance of the *FanRALF3* putative promoter elements idenfied, three progressive truncated fragments of the *FanRALF3-1* upstream sequence were cloned into pKGWFS7 vector and fused to two tandem reporter genes eGFP and β-glucoronidase (GUS) (S1 Fig). Deletions were 200 bp (T4) and 400 bp deletion (T2) (Fig 7A), *Agrobacterium*-mediated transient transformation of white *Fxa* fruits was performed through injection of bacteria transformed with each of the three constructs. Fruits were infected with *C. acutatum* and analyzed for both reporter genes activity at 24 hpi, through quantification of eGFP transcript in qRT-PCR and histochemical GUS assay for β-glucuronidase activity. GUS reporter activity, visualized as blue color of fruits, showed great variability among infected and mock-infected fruits (Fig 7B). Consistent with this, no significant difference was shown in the eGFP transcript level quantified in *C. acutatum* infected versus control fruit (Fig 7C). This is possibly due to the fruit response to *Agrobacterium* itself, independently from the fungal pathogen. Indeed *Agrobacterium* can be perceived as a pathogen by the fruit and stimulate similar responses as *C. acutatum*, including those leading to *FanRALF3* expression. The variability affecting strawberry *Agrobacterium*-mediated transformation depending on technical and environmental conditions has recently been described [76]. It was shown that the expression level of a reporter gene is normally distributed in a population of 30 treated fruits, with huge variation among different fruits. Other important factors affecting agroinfiltration methodology are the quantity of bacteria injected for each fruits, the stage of fruit ripening and the temperature and incubation time after transient transformation. Perhaps for all of these reasons it was not possible to infer the identity of *FanRALF3-1* promoter elements inducible by fungal pathogens such as *C. acutatum* in these experiments. However, the truncated promoter tests were informative. The GUS activity and eGFP expression were measurable and comparable

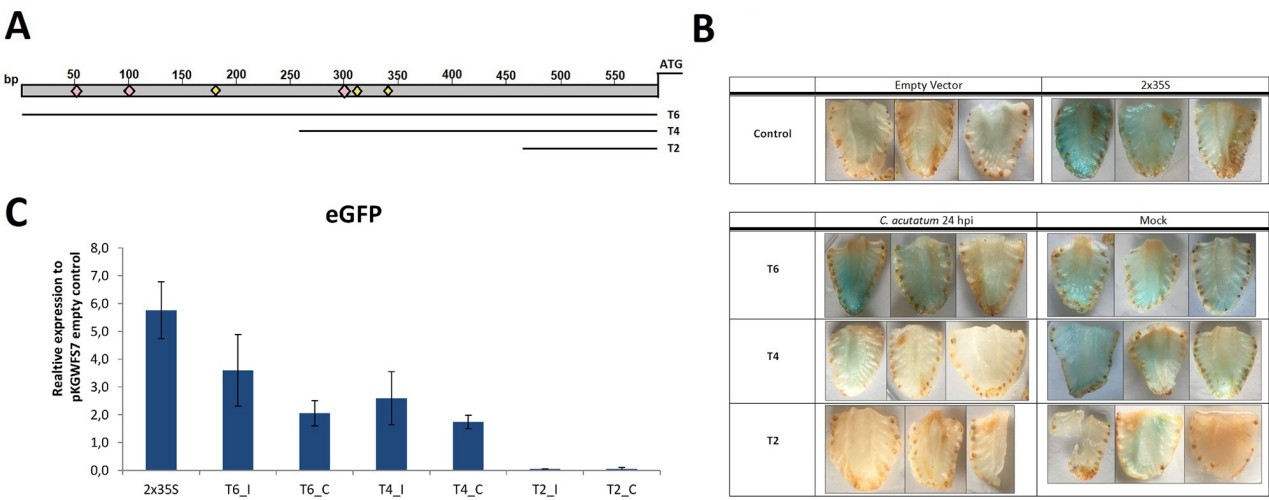

**Fig 7. Dissection of *FanRALF3-1* promoter and *agrobacterium*-mediated reporter assay.** (A) Schematic representation of the promoter fragments used in this study. Pink squares indicate putative MYB-related regulatory elements, yellow squares indicates putative TATA-box related transcriptional activation elements. Histochemical GUS staining to detect β-glucoronidase activity in longitudinal fruit sections of Agroinfiltrated fruits. The table shows fruits transformed with negative (empty vector) and positive (double tandem p35S, 2x35S) controls. Fruits transformed with T6, T4 and T2 truncated promoter fragments were treated with *C. acutatum* or mock-treated and collected 24 h post-infection. (B) Histochemical GUS staining of fruits to detect β-glucoronidase activity in longitudinal fruit sections of Agro-infiltrated fruits. The table shows fruits transformed with negative (empty vector) and positive (double tandem p35S, 2x35S) controls. Fruits transformed with T6, T4 and T2 truncated promoter fragments were treated with *C. acutatum* or mock-treated and collected 24h post-infection. Three replicates for each conitions are shown. (C) Histogram showing qRT- PCR quantitative analysis of eGFP reporter expression. Each bar represents the average of three biological replicates. Expression values were normalized to pKGWFS7 empty vector infiltarted fruits. (C) Mock- tretated samples, (I) *C. acutatum* infected.

from the T4 and T6 sequences, while reporter activity from the T2 element was almost undetectable. These findings suggest that T4, comprising 400 bp sequence upstream *FanRALF3-1* ATG, contains the necessary promoter sequence elements necessary to drive reporter gene expression in strawberry fruits and that the 200 bp region is not sufficient. The full-length upstream region which includes the 3'-UTR of the adjacent gene, shows no more activity that the 400 bp sequence, suggesting this region lacks additional elements to affect transcript accumulation over the T4 construct. According to Motif Scanning analysis, the T4 promoter fragment includes at least two regulatory elements known to be recognized by transcriptional activation complex, (TATA-boxes and Initiator of activation in TATA-less promoter) (Figs 5 and 7A).

## Conclusions

Rapid Alkalinization Factors are small signal peptides with multiple roles in plant growth, fertilization and disease response. RALF genes are upregulated in different plant hosts upon pathogens attack and sequences similar to RALF genes are also expressed by many fungal pathogens as virulence factors, suggesting a role as susceptibility factors during plant pathogen interaction. The present work aimed to characterize the RALF gene family in *F. vesca* woodland and *F.* x *ananassa* octoploid strawberries according to tissue specific expression and similarity to *Arabidopsis* RALFs. The RALF gene family members distribution among *F × ananassa* subgenome is consistent with octoploid genome evolution, which is characterized by different transposable element activity in the different subgenomes and consequently differential gene distribution. A putative involvement of a MYB transcription factor as regulator of *FanRALF3-1* infection-inducibility is speculated based on *in silico* promoter prediction and MYB motif recognition. This element is present in the 400 bp upstream the start codon of *FanRALF3-1* gene sequence. Because *Fan*RALF3-1 contains the same conserved N-terminal sequence as *At*RALF23 it is speculated that *Fan*RALF3-1 interaction with receptor FERONIA (MRLK47) and the coreceptor *Fan*LLG2 may follow the *Arabidopsis* complex interaction structure. Future efforts will seek to identify specific pathogen-responsive promoter elements and their role in strawberry disease resistance.

## Supporting information

**S1 Fig. Map of pKGWFS7 plasmid used for promoter reporter assay.** Sm/Spr, spectinomycin resistance. Kan, kanamycin resistance. FveRALF3 promoter, T6,T4,T2, empty or 2x35S promoter according to experimental procedure. eFGPand GUS, chimera reporter gene formed by eGFP and β-glucoronidase ORFs in frame. T35S, terminator.
(PNG)

**S2 Fig. *F.vesca* RALF peptide sequences aligned in clustal omega.** Black boxes highlight conserved domains RRILA cleavage site, YISY activation site, and conserved cysteines (*). The two proteins coded by *FveRALF15* in different frames were annotated as FveRALF15-F1 (5'-3' Frame1) and FveRALF15-F2 (5'-3' Frame2).
(PNG)

**S3 Fig. FveRALF phylogenetic analysis and clades determination.** Phylogenetic tree shows the classification of *FveRALF* genes in three clades, which were named according to sequence feature similarity with Campbell and Turner clades classification. clade IV (magenta), clade III (light blue) and clade I-II (blue). Neighbour-joining tree values are listed near genes name.
(PNG)

**S4 Fig. *F.vesca* and *Fragaria x ananassa* RALF peptide sequences aligned in clustal omega.** Black boxes highlight conserved domains RRILA cleavage site, YISY activation site, and

conserved cysteines (*).
(PNG)

**S5 Fig. Blastn output of *FanRALF3-1* upstream sequence against *Fragaria x ananassa cv. Camarosa* v1.0.a1 pseudomolecule using GDR.**
(TXT)

**S6 Fig. *FanRALF3-1* putative promoter sequence alignment in different *Fragaria x ananassa* varieties.** It was considered *Fragaria x ananassa cv. Florida Elyana from Florida (U.S.A)*, the Italian variety *cv.Alba* and the sequenced *cv. Camarosa (v1.0.a1)*.
(DOCX)

**S1 Table. List of *Fragaria x ananassa* RALF genes identified through *FveRALFs* BLASTx (GDR).** In the table are listed for each gene, Chromosome localization 'Chr', subgenome localization 'Subg.', gene identification number used to name FanRALF in this work 'geneID', '*Fragaria x ananassa cv. Camarosa* transcript v1.0.a1 annotation', 'Genome location' coordinates, classification in clades according to Campbell and Turner (2017) 'clade', *F. vesca* orthologous gene 'Fv.Ort.', 'E-value' and identity rate 'identity'. 'Gene ID' column presents '*v1short name*' reporting gene v1 annotation abbreviation used to identify genes in Fig 1A and '*orthology based*' nomenclature, assigned according to FveRALF orthology and chromosome lineage: -1, -2,-3,-4 after FveRALF orthologous name were used respectively to indicate *F. vesca*, *F. iinumae*, *F. nipponica* and *F. viridis* progenitors, progressive letters was used to name genes orthologous to the same FveRALF gene.
(XLSX)

**S2 Table. List of primers used for RALFs qRT-PCR expression analysis In *Fragaria x ananassa* infected fruits, and for reporter gene expression in transient transformed fruits.**
(XLSX)

**S3 Table. Motif similarity analysis comparing target motif "S000362" against motifs from "PLACE".** The table shows MotifLab 'motif similarity analysis' output listing: PLACE motif identification code (Motif ID), Motif abbreviated name (Name), Average Log-Likelihood Ratio values (ALLR), Chi-squared values (Chi2), Kullback-Leibler Divergence values (KLD), Pearson's correlation values (Corr), Pearson's correlation weighted values (Corrw), sum of squared distances values (SSD) and consensus sequence (Logo). Motifs are sorted according to ALLR values.
(XLS)

## Acknowledgments

We thank Mrs. Beata Blaszczyk and Dr. Aleksei Lulla for the kind help and support given during the period FN spent at University of Cambridge, UK.

We wish thank Prof. Bruno Mezzetti and his group lab members (Polytechnic University of Marche) for continuous support and suggestions on the experimental work carried on strawberry plants.

## Author Contributions

**Data curation:** Kevin O'Grady, Marko Hyvönen, Kevin M. Folta, Elena Baraldi.

**Funding acquisition:** Elena Baraldi.

**Investigation:** Francesca Negrini, Kevin O'Grady, Marko Hyvönen.

**Methodology:** Francesca Negrini, Kevin O'Grady, Marko Hyvönen, Kevin M. Folta, Elena Baraldi.

**Supervision:** Kevin O'Grady, Marko Hyvönen, Kevin M. Folta, Elena Baraldi.

**Writing – original draft:** Francesca Negrini.

**Writing – review & editing:** Kevin O'Grady, Marko Hyvönen, Elena Baraldi.

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
