## [Decision Letter · Decision Letter 0]

2 Jan 2020

PONE-D-19-32656

Rapid Alkalinization Factor (RALF) gene family genomic structure and transcriptional regulation during host-pathogen crosstalk in Fragaria vescaand Fragariax ananassastrawberry

PLOS ONE

Dear Prof. Baraldi,

Thank you for submitting your manuscript to PLOS ONE. After careful consideration, we feel that it has merit but does not fully meet PLOS ONE’s publication criteria as it currently stands. Therefore, we invite you to submit a revised version of the manuscript that addresses the points raised during the review process.

I agree with both reviewers that this was an interesting study on an important topic. However, a number of concerns were raised. While all of the reviewer's comments should be addressed, most can likely be done so by editing or expanding the text. However, Reviewer 2 noted that more detailed searches for new RALFs are warranted, and this might require additional experimentation. In addition, Reviewer 1 requested more details about Figure 2 and noted difficulties with the quality of this figure.  Reviewer 1 also asked about the 5' upstream sequence of FaRALF6, and I wonder whether some functional analysis of this promoter is possible.

We would appreciate receiving your revised manuscript by Feb 16 2020 11:59PM. To enhance the reproducibility of your results, we recommend that if applicable you deposit your laboratory protocols in protocols.io, where a protocol can be assigned its own identifier (DOI) such that it can be cited independently in the future. For instructions see: http://journals.plos.org/plosone/s/submission-guidelines#loc-laboratory-protocols

We look forward to receiving your revised manuscript.

Kind regards,

Richard A Wilson

Academic Editor

PLOS ONE

Reviewers' comments:

Reviewer's Responses to Questions

**Comments to the Author**

1. Is the manuscript technically sound, and do the data support the conclusions?

Reviewer #1: Yes

Reviewer #2: Partly

2. Has the statistical analysis been performed appropriately and rigorously? 

Reviewer #1: Yes

Reviewer #2: I Don't Know

3. Have the authors made all data underlying the findings in their manuscript fully available?

Reviewer #1: Yes

Reviewer #2: Yes

4. Is the manuscript presented in an intelligible fashion and written in standard English?

Reviewer #1: Yes

Reviewer #2: Yes

5. Review Comments to the Author

Reviewer #1: Title: “Rapid Alkalinization Factor (RALF) gene family genomic structure and transcriptional regulation during host-pathogen crosstalk in Fragaria vesca and Fragaria x ananassa strawberry”

Authors: Francesca Negrini, Kevin O’Grady, Marko Hyvönen, Kevin M. Folta and Elena Baraldi.

General Comments:

The paper consists in the identification, characterization and genomic organization of the RALF gene family of F. x ananassa and F. vesca strawberry species based on the information available of Arabidopsis ortholog genes. Results revealed that whereas F. vesca contains 13 RALF genes, F. x ananassa contains 50, and they do not show conserved localization among the genotypes analyzed. Evaluation of the effect of infection with the fungal pathogens C. acutatum or B. cinerea on the expression of 7 selected FaRAFL genes in strawberry fruits showed that only FaRALF3 was upregulated, and FaRALF6 downregulated after fungal infection. The paper also includes the characterization of the 5’ upstream sequence of the gene FaRALF3, corresponding to the putative promoter region, and the evaluation of its functionality when fruits were infected with the pathogens C. acutatum or B. cinerea. Experiments were carried out by agroinfiltrating strawberry fruit with FaRALF3 truncated 5’ upstream sequences and evaluating the level of expression of GFP and GUS genes in fruits infected with C. acutatum or B. cinerea.

The paper is interesting and provides important information about the structure, genomic arrangement and transcriptomic of the RALF genes of strawberry species.

Overall, my opinion is that the paper deserves to be published after minor revision and answering the comments pointed out below.

Specific Comments

1) Line 63. Revise what “Buddah’s Paper Seal 1, 2” means.

2) Line 117. Revise, “..to gaining...” . Should be “..to gain...” or “…of gaining…”.

3) Lines 153-159. When fruits were inoculated with the pathogens, they were evaluated 24 hpi. However, whereas for B. cinerea that time may be adequate for assessing the effect on RALF expression, that time may not be adequate for C. acutatum. Normally the latter requires more time; likewise can be discussed about the optimal temperatures required for each pathogen. These experimental details may have consequences when evaluating transcriptional levels of RALF genes. I suggest that authors should add some comments about this issue.

4) In Figure 2, authors show a complex figure in which they present a transcriptional analysis of RALD genes in F. vesca. The figure I have received is not quite clear, therefore I could not see clearly plants stages. Anyhow authors should explain better in the text or in the figure legend the stages indicated at the botton of this figure. Since the transcription level of all genes evaluated may change with time, authors should be more precise about the physiological state of plants used, as they did with fruits. They should more details about how were the plants grown (i.e. photosynthetic photon flux density used), and how old were the plants, the leaves, and roots at the moment of taking the samples? What roots did they use for the analysis (e.g. whole root, principal, lateral)? Likewise with the leaves. Did they take the three leaflets, or only the central one for the analysis?

5) Line 413. Revise. There is a “t” that should be deleted.

6) Line 419. Revise. “...since the F.ve progenitor...”. What is F.ve? F. vesca?

7) It is interesting the study carried out by the authors of the 5’ upstream sequence of the gene FaRALF3, and understand why did they focused on that gene, however I wonder why did not pay equal attention to the 5’ upstream sequence of the gene FaRALF6. These genes exhibit an opposite behavior when infected with C. acutatum or B. cinerea, mostly in red fruits. Do they share similar cis-regulatory sequences, or cis-acting elements? It would be desirable if the authors add some comment about it.

Reviewer #2: As a qualification to my reply to the prior question (about standard English), the manuscript could benefit by careful editing by a native English speaker. While the writing is clear and unambiguous, there are some systematic departures from standard English in the use, or lack of use, of articles such as "the", and also distinction between singular/plural forms.

That minor detail aside, the manuscript content is generally interesting and useful, but in my view a number of issues would need to be addressed to warrant publication. These issues are described, and other minor editorial suggestions are provided, on a line-by-line basis below.

Line 21: Replace “Factor” with “Factors”

Line 21: Why are RALFs referred to here and throughout the manuscript as peptides rather than polypeptides or proteins? What criteria distinguish these three terms/classes and on what basis is the term “peptide” utilized in this manuscript? This needn’t be explained in the Abstract, but it should be addressed in the Introduction. Once the distinction is explained, consistent terminology should be used throughout the manuscript.

Line 23: insert “,” after “development”

Line 23: replace “regulator” with “regulators”, and check throughout manuscript for proper usage of singular/plural.

Line 25: are incomplete citations such as this permitted in the Abstract in PLOS-ONE?

Line 28: delete “type of” (in what sense are these two varieties “types”?)

Line 29: insert comma after “expression”

Line 29: specific F. vesca cultivar/accession: it may not be typical.

Line 30: specific F. x ananassa cultivar: it may not be typical.

Line 31: it has not yet been established that F. x ananassa has four definable subgenomes, or that the subgenome composition of variety “Camaroso” is typical or representative of F. x ananassa.

Lne 31: the statement about transposable elements should be deleted: the manuscript provides no data on TE distributions in the studied species.

Line 32: replaces “genes” with “gene”

Line 33: state pathogen genus names on first usage.

Lines 36 and 38: should the gene name end in -1?

Line 47: replace “to respond” with “that respond”.

Line 48: replace “organs” with “organ”

Line 50: replace “in plant” with “in the plant”

Line 63: spelling error in protein name

Line 65: What is “pollen the tube”?

Lines 92-95: cite information sources.

Line 115: it has not yet been established that F. x ananassa cultivars have four, consistently definable subgenomes, or that “Camarosa” is typical. No other studies have defined octoploid subgenome composition in the same way. When referring to the subgenome model for Camarosa published by Edger et al., 2019, it would be advisable to qualify your statements to make it clear that their model is based on one study of one cultivar, and that it is still best regarded as “tentative” or “putative”.

Similarly, statements made about F. vesca Hawaii 4 (FvH4) should be qualified to limit conclusions specifically to this highly atypical representative of F. vesca, which in its wild type form has red fruit and short-day photoperiodic response as opposed to the yellow fruit and day neutrality of H4.

Line 130: Compared to what, and integrated into what? Not clear.

Line 140: Run a spell-check here and throughout manuscript.

Lines 258-260. It does not appear that the authors used a Blast search of the F. vesca genome, but relied only of the effectiveness of one search term “RALF”. They found 13 genes that had been previously identified as RLAF genes in the process of annotation by prior investigations. Thus, it seems unjustified for the authors to state that they have “identified” these 13 genes. They simply used a word search for find those RALF genes previously identified by others. Also, this word search approach is in my view insufficient for the intended purpose, which was to discover new RALF genes,, especially given that the investigators later used the 13 F. vesca RALF genes as queries in Blast searches of the octoploid Fxa genome. Why not also search back through to vesca genome using the 13 genes as queries to see if anything had been missed by the published genome annotation? This would be easy to do, and I would consider it a requirement in the revision.

Line 294: define “Fxa” upon first use.

Lines 351-352. Some additional justification is needed for the choice of “available RNA-seq datasets” in the performance of comparative expression analyses in F. vesca. Seemingly, the original purpose of these datasets was to assist in gene annotation of the F. vesca genome by other investigators. But what makes this set of sequence suitable for comparative expression analysis? Please explain.

Line 494: the examined upstream region was only 400 bp in length, which seems very small to warrant the term “putative promoter”. It is quite possible that interesting upstream polymorphisms would have been discovered by sequencing further upstream, on which basis it is not justifiable to state, as the authors have done, that “susceptibility to anthracnose disease cannot be associated with allelic polymorphisms”. It would appear accurate to say that in the limited sequence region examined, no association was detected.

6. PLOS authors have the option to publish the peer review history of their article (what does this mean?). If published, this will include your full peer review and any attached files.

Reviewer #1: No

Reviewer #2: No

---

## [Author Response · Author response to Decision Letter 0]

18 Feb 2020

Thank you for submitting your manuscript to PLOS ONE. After careful consideration, we feel that it has merit but does not fully meet PLOS ONE’s publication criteria as it currently stands. Therefore, we invite you to submit a revised version of the manuscript that addresses the points raised during the review process.

I agree with both reviewers that this was an interesting study on an important topic. However, a number of concerns were raised. While all of the reviewer's comments should be addressed, most can likely be done so by editing or expanding the text. However, Reviewer 2 noted that more detailed searches for new RALFs are warranted, and this might require additional experimentation. In addition, Reviewer 1 requested more details about Figure 2 and noted difficulties with the quality of this figure. Reviewer 1 also asked about the 5' upstream sequence of FaRALF6, and I wonder whether some functional analysis of this promoter is possible.

We thank the editor and referees for the comments and suggestions that allowed us to improve our work. A new MS version addressing all the criticism (including English language revision, a more extended analysis and search of RALF genes, Figure 2 improvement and RALF6 promoter analysis), was prepared, hoping to meet now the PLOS ONE standard requirements. 

Reviewer #1: Title: “Rapid Alkalinization Factor (RALF) gene family genomic structure and transcriptional regulation during host-pathogen crosstalk in Fragaria vesca and Fragaria x ananassa strawberry”

Authors: Francesca Negrini, Kevin O’Grady, Marko Hyvönen, Kevin M. Folta and Elena Baraldi.

General Comments:

The paper consists in the identification, characterization and genomic organization of the RALF gene family of F. x ananassa and F. vesca strawberry species based on the information available of Arabidopsis ortholog genes. Results revealed that whereas F. vesca contains 13 RALF genes, F. x ananassa contains 50, and they do not show conserved localization among the genotypes analyzed. Evaluation of the effect of infection with the fungal pathogens C. acutatum or B. cinerea on the expression of 7 selected FaRAFL genes in strawberry fruits showed that only FaRALF3 was upregulated, and FaRALF6 downregulated after fungal infection. The paper also includes the characterization of the 5’ upstream sequence of the gene FaRALF3, corresponding to the putative promoter region, and the evaluation of its functionality when fruits were infected with the pathogens C. acutatum or B. cinerea. Experiments were carried out by agroinfiltrating strawberry fruit with FaRALF3 truncated 5’ upstream sequences and evaluating the level of expression of GFP and GUS genes in fruits infected with C. acutatum or B. cinerea.

The paper is interesting and provides important information about the structure, genomic arrangement and transcriptomic of the RALF genes of strawberry species.

Overall, my opinion is that the paper deserves to be published after minor revision and answering the comments pointed out below.

1) Line 63. Revise what “Buddah’s Paper Seal 1, 2” means. 

It was added the term ‘proteins’ to the name ‘Buddha’s Paper Seal 1, 2’. Now line 62-63.

As it is described in the cited literature [13] :

Ge Z, Bergonci T, Zhao Y, Zou Y, Du S, Liu MC, et al. Arabidopsis pollen tube integrity and sperm release are regulated by RALF-mediated signaling. Science (80- ). 2017; doi: 10.1126/science.aao3642

the name ‘Buddha’s Paper Seal 1, 2’ were given to the genes At4g39110 and At2g21480 based on the pollen tube phenotypes of their knockout mutants, since, as reported in the paper, ‘The precocious pollen tube bursting phenotype in bups mutants reminded us of the famous Chinese myth “Journey to the West,” where the Monkey King was once trapped under a mountain and sealed by a Buddha’s Paper Seal for 500 years before a monk heading to the West removed the seal to release him. The two receptors are therefore named BUPSs.’

2) Line 117. Revise, “..to gaining...” . Should be “..to gain...” or “…of gaining…”.

done.

3) Lines 153-159. When fruits were inoculated with the pathogens, they were evaluated 24 hpi. However, whereas for B. cinerea that time may be adequate for assessing the effect on RALF expression, that time may not be adequate for C. acutatum. Normally the latter requires more time; likewise can be discussed about the optimal temperatures required for each pathogen. These experimental details may have consequences when evaluating transcriptional levels of RALF genes. I suggest that authors should add some comments about this issue.

The time 24 hpi time was chosen since in our previous studies (reference 29: Guidarelli M, Carbone F, Mourgues F, Perrotta G, Rosati C, Bertolini P, et al. Colletotrichum acutatum interactions with unripe and ripe strawberry fruits and differential responses at histological and transcriptional levels. Plant Pathol. 2011;60(4):685–97) we showed that C. acutatum infecting white fruits became quiescent as melanized appressoria after 24 hpi, whereas active subcuticular necrotrophic invasion was displayed on red fruits. Furthermore FaRALF3 was found overexpressed in red fruits after 24 hours post infection suggesting that it could play a role as susceptibility gene. 

With respect to temperature, we chose to infect at room temperature (about 21°C) since this condition was the one used also in previous experiments (same ref as above, Guidarelli et al 2011) where RALF3 upregulation was firstly highlighted.

Now these conditions are specified (line 154-155).

4) In Figure 2, authors show a complex figure in which they present a transcriptional analysis of RALD genes in F. vesca. The figure I have received is not quite clear, therefore I could not see clearly plants stages. Anyhow authors should explain better in the text or in the figure legend the stages indicated at the botton of this figure. Since the transcription level of all genes evaluated may change with time, authors should be more precise about the physiological state of plants used, as they did with fruits. They should more details about how were the plants grown (i.e. photosynthetic photon flux density used), and how old were the plants, the leaves, and roots at the moment of taking the samples? What roots did they use for the analysis (e.g. whole root, principal, lateral)? Likewise with the leaves. Did they take the three leaflets, or only the central one for the analysis?

The RALF genes transcriptional information reported in figure 2 were obtained by elaborating available transcriptome datasets [ref. 34], therefore the details about growth stages and sampling conditions are reported in the articles reporting those original data. Maybe this was not clear enough in our previous version of the paper. We hope now to have made it clearer by adding details about plant developmental stages in the figure 2 legend (line 423-449) and more references in the material and methods section (line 142). Figure 2 image quality was improved.

5) Line 413. Revise. There is a “t” that should be deleted. Done.

6) Line 419. Revise. “...since the F.ve progenitor...”. What is F.ve? F. vesca? Done.

7) It is interesting the study carried out by the authors of the 5’ upstream sequence of the gene FaRALF3, and understand why did they focused on that gene, however I wonder why did not pay equal attention to the 5’ upstream sequence of the gene FaRALF6. These genes exhibit an opposite behavior when infected with C. acutatum or B. cinerea, mostly in red fruits. Do they share similar cis-regulatory sequences, or cis-acting elements? It would be desirable if the authors add some comment about it.

We thank the referee for this nice suggestion. We have followed this advice and made the prediction analysis of pathogen-responsive regulatory elements also on FaRALF6-1 promoter. A new figure and supplemental table (Fig 6 and Table S3) and a new description of FaRALF6 motif scanning and discovery analysis was added (line 614), do to reply to referee’s question (Do they share similar cis-regulatory sequences, or cis-acting elements?) 

 

Reviewer #2:

 As a qualification to my reply to the prior question, the manuscript could benefit by careful editing by a native English speaker. While the writing is clear and unambiguous, there are some systematic departures from standard English in the use, or lack of use, of articles such as "the", and also distinction between singular/plural forms. 

That minor detail aside, the manuscript content is generally interesting and useful, but in my view a number of issues would need to be addressed to warrant publication. These issues are described, and other minor editorial suggestions are provided, on a line-by-line basis below.

Line 21: Replace “Factor” with “Factors” done

Line 21: Why are RALFs referred to here and throughout the manuscript as peptides rather than polypeptides or proteins? What criteria distinguish these three terms/classes and on what basis is the term “peptide” utilized in this manuscript? This needn’t be explained in the Abstract, but it should be addressed in the Introduction. Once the distinction is explained, consistent terminology should be used throughout the manuscript.

We thank the referee for this indication. Indeed, the term ‘peptides’ refers to molecules ranging from 2 to 50 amino acids, whereas ‘polipeptides’ or proteins’ are made of 50 or more amino acids. Since the RALF genes code for proteins with a full length of 80–120 amino acids, but the active form originating from protease cleavage, consists of about 50 aminoacids (now stated at line 51), RALFs are named as peptides when the context refers to their active form and as proteins when refers to the family. The manuscript was revised using the two terms properly according to the context. 

Line 23: insert “,” after “development” Done

Line 23: replace “regulator” with “regulators”, and check throughout manuscript for proper usage of singular/plural. Done

Line 25: are incomplete citations such as this permitted in the Abstract in PLOS-ONE? Citation was removed 

Line 28: delete “type of” (in what sense are these two varieties “types”?) Done

Line 29: insert comma after “expression” Done

Line 29: specific F. vesca cultivar/accession: it may not be typical. Done

Line 30: specific F. x ananassa cultivar: it may not be typical. Done

Line 31: it has not yet been established that F. x ananassa has four definable subgenomes, or that the subgenome composition of variety “Camaroso” is typical or representative of F. x ananassa. 

We have revised the manuscript adding a line (324-327) which specifies that the speculation is limited to the Fragaria x ananassa cv. Camarosa.

Lne 31: the statement about transposable elements should be deleted: the manuscript provides no data on TE distributions in the studied species. Done

Line 32: replaces “genes” with “gene” Done

Line 33: state pathogen genus names on first usage. Done

Lines 36 and 38: should the gene name end in -1? yes, revised

Line 47: replace “to respond” with “that respond”. Done

Line 48: replace “organs” with “organ” ok Done 

Line 50: replace “in plant” with “in the plant”Done

Line 63: spelling error in protein name Corrected

Line 65: What is “pollen the tube”? Corrected 

Lines 92-95: cite information sources. Done

Line 115: it has not yet been established that F. x ananassa cultivars have four, consistently definable subgenomes, or that “Camarosa” is typical. No other studies have defined octoploid subgenome composition in the same way. When referring to the subgenome model for Camarosa published by Edger et al., 2019, it would be advisable to qualify your statements to make it clear that their model is based on one study of one cultivar, and that it is still best regarded as “tentative” or “putative”.

Similarly, statements made about F. vesca Hawaii 4 (FvH4) should be qualified to limit conclusions specifically to this highly atypical representative of F. vesca, which in its wild type form has red fruit and short-day photoperiodic response as opposed to the yellow fruit and day neutrality of H4.

As suggested by the referee, we specified that the subgenomic organization is a prediction (line 310-311) and that our analysis is limited to the observation of one cultivar (324-327). 

Line 130: Compared to what, and integrated into what? Not clear. The sentence has been rephrased. Line 124-127.

Line 140: Run a spell-check here and throughout manuscript. done.

Lines 258-260. It does not appear that the authors used a Blast search of the F. vesca genome, but relied only of the effectiveness of one search term “RALF”. They found 13 genes that had been previously identified as RLAF genes in the process of annotation by prior investigations. Thus, it seems unjustified for the authors to state that they have “identified” these 13 genes. They simply used a word search for find those RALF genes previously identified by others. Also, this word search approach is in my view insufficient for the intended purpose, which was to discover new RALF genes,, especially given that the investigators later used the 13 F. vesca RALF genes as queries in Blast searches of the octoploid Fxa genome. Why not also search back through to vesca genome using the 13 genes as queries to see if anything had been missed by the published genome annotation? This would be easy to do, and I would consider it a requirement in the revision.

We agree with the referee on this point. Indeed, in our previous version, we had performed also a Blastn search using the genes previously found trough the keyword analysis as query against F. vesca v4.0.a2 transcriptome. However, we decided to not include the results in the manuscript since the genes found did not show high similarity to RALF typical conserved features. In any case, as referee suggested, we performed again the analysis using, this time, the F. vesca chromosomes dataset instead of transcriptome. We found two other putative RALF genes that we have now included in Table 1 and described from line 271 to 281. 

RALF Clades description (line 282-287), figure 1, Table S1, Figure S2 and S4 have been modified accordingly, and a new Figure S3 prepared.

Line 294: define “Fxa” upon first use. Done in line 318.

Lines 351-352. Some additional justification is needed for the choice of “available RNA-seq datasets” in the performance of comparative expression analyses in F. vesca. Seemingly, the original purpose of these datasets was to assist in gene annotation of the F. vesca genome by other investigators. But what makes this set of sequence suitable for comparative expression analysis? Please explain.

As far as we know, the original purpose of the transcriptome datasets, used by Li et al. (2019) for strawberry genome reannotation, was to analyse genes transcriptional regulation during respectively flower and fruit development, roots infection with Phytophthora cactorum and vegetative tissues growth (seedlings and leaves). These studies were conducted respectively by Kang et al.(2013), Hollender et al., (2014) and Toljamo et al., (2016). Thus the manuscript was revised adding the proper references at line 146 when RNAseq datasets were described.

Line 494: the examined upstream region was only 400 bp in length (which seems very small to warrant the term “putative promoter” (It is quite possible that interesting upstream polymorphisms would have been discovered by sequencing further upstream, on which basis it is not justifiable to state, as the authors have done, that “susceptibility to anthracnose disease cannot be associated with allelic polymorphisms”. It would appear accurate to say that in the limited sequence region examined, no association was detected.

We analysed a region of 656 bp which, according to F. vesca v4.0.a2 genes annotation version, corresponds to the non coding region between FvRALF3 ATG and the STOP codon of the upstream flanking gene (gene FvH4_2g25340) of FvRALF3 (lines 522-524). Indeed it is possible that polymorphisms in other regions, even very far from the region analyzed, are involved in FvRALF3 gene transcriptional regulation and susceptibility to C. acutatum. Now this consideration is stated in line 538-539.

---

## [Decision Letter · Decision Letter 1]

2 Mar 2020

Genomic structure and transcript analysis of the Rapid Alkalinization Factor (RALF) gene family during host-pathogen crosstalk in Fragaria vesca and Fragaria x ananassa strawberry

PONE-D-19-32656R1

Dear Dr. Baraldi,

We are pleased to inform you that your manuscript has been judged scientifically suitable for publication and will be formally accepted for publication once it complies with all outstanding technical requirements.

Please note that Reviewer 1 has recommended a minor revision, but this can likely be addressed during the proofing stage.

With kind regards,

Richard A Wilson

Academic Editor

PLOS ONE

Additional Editor Comments (optional):

Reviewers' comments:

Reviewer's Responses to Questions

**Comments to the Author**

1. If the authors have adequately addressed your comments raised in a previous round of review and you feel that this manuscript is now acceptable for publication, you may indicate that here to bypass the “Comments to the Author” section, enter your conflict of interest statement in the “Confidential to Editor” section, and submit your "Accept" recommendation.

Reviewer #1: All comments have been addressed

Reviewer #2: All comments have been addressed

2. Is the manuscript technically sound, and do the data support the conclusions?

Reviewer #1: Yes

Reviewer #2: Yes

3. Has the statistical analysis been performed appropriately and rigorously? 

Reviewer #1: Yes

Reviewer #2: N/A

4. Have the authors made all data underlying the findings in their manuscript fully available?

Reviewer #1: Yes

Reviewer #2: Yes

5. Is the manuscript presented in an intelligible fashion and written in standard English?

Reviewer #1: (No Response)

Reviewer #2: Yes

6. Review Comments to the Author

Reviewer #1: Authors have answered all the queries and comments I have formulated, and completed satisfactorily those aspects that were pointed out in the previous version. They have also incorporated a new figure (Fig. 6) and a new Table (Table S3) to analyze the 5’ upstream, as suggested.

However, there is still a minor comment I would suggest the authors to incorporate.

In Figure 3 there is no indication that the gene expression levels shown are relative to the reference gene Elongation Factor 1. The latter can be done by adding at the figure caption a line saying that the numbers correspond to the “relative expression level”, or at the ordinate.

As far as I am concerned, my opinion is that the paper can be published after incorporating the above mentioned comment.

Reviewer #2: (No Response)

7. PLOS authors have the option to publish the peer review history of their article (what does this mean?). If published, this will include your full peer review and any attached files.

Reviewer #1: No

Reviewer #2: No

---

## [Editor Report · Acceptance letter]

6 Mar 2020

PONE-D-19-32656R1 

Genomic structure and transcript analysis of the Rapid Alkalinization Factor (RALF) gene family during host-pathogen crosstalk in *Fragaria vesca* and *Fragaria* x *ananassa* strawberry 

Dear Dr. Baraldi:

I am pleased to inform you that your manuscript has been deemed suitable for publication in PLOS ONE. Congratulations! Your manuscript is now with our production department. 

With kind regards,

on behalf of

Dr. Richard A Wilson 

Academic Editor

PLOS ONE